# Free Vibration Responses of Functionally Graded CNT-Reinforced Composite Conical Shell Panels

**DOI:** 10.3390/polym15091987

**Published:** 2023-04-22

**Authors:** Jin-Rae Cho

**Affiliations:** Department of Naval Architecture and Ocean Engineering, Hongik University, Jochiwon, Sejong 30016, Republic of Korea; jrcho@hongik.ac.kr; Tel.: +82-44-860-2546

**Keywords:** FG-CNTRCs, conical shell panels, free vibration responses, 2-D natural element method (NEM), MITC3+shell element, parametric investigation

## Abstract

Functionally graded CNT (carbon nanotube)-reinforced composites (FG-CNTRCs) are intensively studied because the mechanical behaviors of conventional composites can be dramatically improved. Only a small amount of CNTs are appropriately distributed through the thickness. However, the studies on conical shell panels have been poorly reported when compared with beams, plates and cylindrical shells, even though more parameters are associated with the mechanical behaviors of conical shell panels. In this context, this study intends to profoundly investigate the free vibration of FG-CNTRC conical shell panels by developing an effective and reliable 2-D (two-dimensional) numerical method. The displacement field is expressed using the first-order shear deformation shell theory, and it is approximated by the 2-D planar natural element method (NEM). The conical shell surface is transformed into the 2-D planar NEM grid, and the approach for MITC3+shell element is employed to suppress the shear locking. The developed numerical method is validated through the benchmark experiments, and the free vibration responses of FG-CNTRC conical shell panels are investigated with respect to all the associated parameters. It is found from the numerical results that the free vibration of FG-CNTRC conical shell panels is significantly influenced by the volume fraction and distribution pattern of CNTs, the geometry parameters of the conical shell, and the boundary condition.

## 1. Introduction

Carbon nanotube (CNT) has attracted great attention as a promising material for the 21st century thanks to its excellent physical and chemical properties, such as its high mass-strength ratio. Since its introduction, it has been widely adopted as a next-generation pillar for polymer composites because the addition of a small amount of CNTs dramatically increases the stiffness of CNT-reinforced composites called CNTRC [1,2]. For use as mechanical and structural members, CNTRCs are usually developed in the forms of beams, plates and shells. Later, the thickness-wise distribution pattern of CNTs was purposefully assumed [1,3] according to the concept of functionally graded materials (FGMs) [4]. In FGMs, the thickness-wise volume fractions of base materials vary continuously and may be optimally tailored so as to maximize the target function [5]. The CNTRCs with functionally graded CNT distributions through the thickness are called FG-CNTRCs, and several representative ones, such as FG-U, FG-V, FG-O, FG-X and FG-Λ, were introduced. The purpose of these FG-CNTRCs was to allow one to choose one among them which is most suitable for the target performance, and it requires a profound investigation of the mechanical characteristics of each FG-CNTRC. In this context, extensive research efforts have focused on the investigation of mechanical responses such as static bending, free vibration and buckling. The research works on CNTRCs and FG-CNTRCs would be referred to the references [6,7,8].

The studies on FG-CNTRCs have been made analytically using the shear deformation-based theories or numerically using the finite element method. However, these studies have been mostly concerned with beams, plates and cylindrical shells, so that studies on conical shells have been rarely reported. When compared with beam, plate and cylindrical shell, the mechanical behaviors of the conical shell are influenced by more parameters such as semi-vertex angle and sub-tended angle as well as radius, thickness, width and length. According to the literature survey, Nguyen Dinh and Nguyen [9] numerically investigated the dynamic response of FG-CNTRC truncated conical shells resting on elastic foundations using the Galerkin method and the fourth-order Runge–Kutta method. Kiani et al. [10] analyzed the natural frequencies of FG-CNTRC conical shell panels using the first-order shear deformation shell theory (FSDT) and Donnell’s theory. Sofiyev et al. [11] presented an effective analytical solution for the stability problem of FG-CNTRC conical shells exposed to external lateral and hydrostatic pressure. Chan et al. [12] numerically investigated the nonlinear buckling and post-buckling responses of FG-CNTRC truncated conical shells subject to axial load. Ansari et al. [13] investigated the nonlinear vibration response of FG-CNTRC conical shells using the higher-order shear deformation theory (HSDT) and von-Karman geometric nonlinearity. Qin et al. [14] presented a unified Fourier series solution to solve the free vibration of FG-CNTRC conical shells using FSDT in conjunction with the modified Fourier series and Ritz method. Talebitooti et al. [15] investigated the frequency behavior of the joined conical–conical panels with functionally graded CNT reinforcement using FSDT and Hamilton’s principle. Fu et al. [16] presented an accurate analytical method for investigating the dynamic instability of a laminated FG-CNTRC conical shell surrounded by an elastic foundation. Xiang et al. [17] presented the free vibration analysis of FG-CNTRC conical shell panels using the element-free kernel particle Ritz method to investigate the free vibration characteristics. Hou et al. [18] numerically investigated the free vibration of FG-CNTRC conical shells with an intermediate ring for various boundary conditions. Rezaiee-Pajand et al. [19] predicted the natural frequencies of functionally graded conical shell structures using FSDT and the generalized differential quadrature method (GDQM).

Meanwhile, the MITC (Mixed-interpolated Tensorial Components) approach has been widely adopted for the numerical studies of shell structures to alleviate the locking phenomenon. Lee et al. [20] proposed an effective new 3-node triangle shell finite element called the MITC3+element to reduce shear locking in the shell element. Lyly et al. [21] introduced a new quadrilateral element using isoparametric bilinear basis functions for both rotation and deflection components in order to further stabilize the MITC4 element. Rejaiee-Pajand et al. [22] presented an efficient six-node triangular element by employing the MITC approach to numerically analyze nonlinear static and free vibration of uniform distributed CNTRC structures under in-plane loading. Tran et a. [23] adopted the MITC approach to an edge-based smoothed finite element method (ES-FEM) to numerically investigate static bending and free vibration of functionally graded porous variable-thickness plates. Chau-Dinh [24] presented an enhanced 3-node triangle flat shell element in which new cell-based smoothed bending strains are derived, and the shear locking is suppressed by employing the MITC approach.

It can be realized from the literature survey that the free vibration response of FG-CNTRC conical shell panels has not been sufficiently investigated, particularly the parametric investigation with respect to the major parameters. In this situation, the main goal of this study is to profoundly investigate the free vibration responses of FG-CNTRC conical shell panels by developing a 2-D locking-free reliable and effective numerical method. The numerical method is developed by integrating the 2-D planar natural element method (NEM) and the MITC approach. NEM is a lastly introduced mesh-free method characterized by high smooth Laplace interpolation functions [25,26,27]. The first-order shear deformation shell theory is used to express the displacement field, and the geometry transformation is introduced between the physical conical shell surface and the computational 2-D planar NEM grid. The approach for MITC3+shell finite element [20] is employed to suppress the troublesome shear locking for the bending-dominated thin structures [28]. The developed NEM-based numerical method is verified through the benchmark experiments, and the free vibration responses of FG-CNTRC conical shell panels are investigated with respect to the CNT-associated parameters, the conical shell parameters, the sandwich core thickness, and the boundary conditions.

This paper is organized as follows. Following the introduction, the FG-CNTRC conical shell panel and its displacement, strain and stress fields are addressed in Section 2. The natural element approximation of free vibration of FG-CNTRC conical shell panels is explained in Section 3. The numerical results of benchmark and parametric experiments are presented and discussed in Section 4, and the final conclusion is given in Section.

## 2. Modeling of FG-CNTRC Conical Shell Panel

Figure 1a represents a circular conical shell panel reinforced with single-walled carbon nanotubes (SWCNTs). A coordinate system $\left(x,\theta ,z\right)$ is positioned on the mid-surface of the panel, and the geometric configuration of the conical shell panel is characterized by semi-vertex angle α, sub-tended angle $\theta_{0}$, length ℓ, small radius $R_{0}$ and thickness h. The radius $R\left(x\right)$ of conical shell panel at any point along its length is determined by
(1)$$R\left(x\right)=R_{0}+xsin\;\alpha$$

CNTs are aligned along the x−axis with a specific functionally graded distribution pattern through the thickness. Four representative distribution patterns of CNTs are represented in Figure 1b, where CNTs are uniformly distributed in FG-U while CNTs are rich at the mid-surface in FG-O, rich at the outer shell in FG-X and rich at the inner shell in FG-Λ, respectively. The volume fractions of CNTs and underlying matrix are denoted by $V_{CNT}\left(z\right)$ and $V_{m}\left(z\right)$ which are in the relation given by
(2)$$V_{CNT}\left(z\right)+V_{m}\left(z\right)=1$$
where the CNT volume fractions $V_{CNT}\left(z\right)$ corresponding to the four representative CNT distribution patterns are expressed as
(3)$$V_{CNT}\left(z\right)=\left\{\begin{array}{ll} V_{CNT}^{*}, & \mathrm{FG}-U\\ 2\left(1-2\left|z\right|/h\right)V_{CNT}^{*}, & \mathrm{FG}-O\;\\ 2\left(2\left|z\right|/h\right)V_{CNT}^{*}, & \mathrm{FG}-X\\ \left(1-2z/h\right)V_{CNT}^{*}, & \mathrm{FG}-\Lambda \end{array}\right.$$
with
(4)$$V_{CNT}^{*}=\frac{w_{CNT}}{w_{CNT}+\left(\rho^{CNT}/\rho^{m}\right)-\left(\rho^{CNT}/\rho^{m}\right)w_{CNT}}$$
Here, $w_{CNT}$ is the mass fraction of CNTs occupied within a unit volume of a conical shell panel, and $\rho^{CNT}$ and $\rho^{m}$ denote the CNT density and the matrix density.

As a dual-phase composite material, FG-CNTRC structures are usually modeled as an orthotropic material, and their effective elastic properties are evaluated using the homogenization methods such as the liner rule of mixtures by introducing the direction-wise CNT efficiency parameters $\eta_{j}\left(j=1,2,3\right)$. The effective elastic and shear moduli of the FG-CNTRC conical shell panel are calculated according to [3]:(5)$$E_{11}=\eta_{1}V_{CNT}E_{11}^{CNT}+V_{m}E^{m}$$
(6)$$\frac{\eta_{2}}{E_{22}}=\frac{V_{CNT}}{E_{22}^{CNT}}+\frac{V_{m}}{E^{m}}$$
(7)$$\frac{\eta_{3}}{G_{12}}=\frac{V_{CNT}}{G_{12}^{CNT}}+\frac{V_{m}}{G^{m}}$$
It is assumed that $G_{13}=G_{12}$ and $G_{23}=1.2G_{12}$. Meanwhile, the effective density ρ and the effective Poisson’s ratio $\nu_{12}$ through the thickness are assumed as
(8)$$\rho =V_{CNT}\rho_{}^{CNT}+V_{m}\rho_{}^{m}$$
(9)$$\nu_{12}=V_{CNT}\nu_{12}^{CNT}+V_{m}\nu_{12}^{m}$$

**Figure 1 polymers-15-01987-f001:**
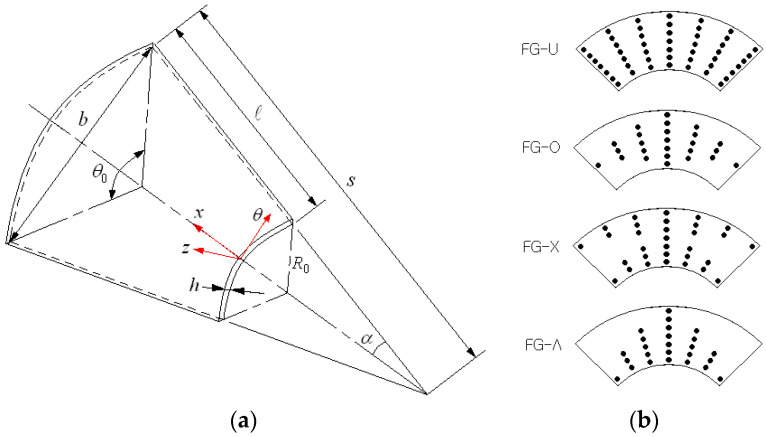
FG-CNTRC conical shell panel: (**a**) the geometry dimensions, (**b**) functionally graded distribution patterns of CNTs.

By adopting the first-order shear deformation shell theory, the displacement field $u={\left\{u,v,w\right\}}^{T}$ is expressed as
(10)$${\left\{\begin{array}{c} u\\ v\\ w \end{array}\right\}}_{\left(x,\theta ,z\right)}={\left\{\begin{array}{c} u_{0}\\ v_{0}\\ w_{0} \end{array}\right\}}_{\left(x,\theta \right)}+z\cdot {\left\{\begin{array}{c} \vartheta_{x}\\ \vartheta_{y}\\ 0 \end{array}\right\}}_{\left(x,\theta \right)}$$
with $d={\left(u_{0},v_{0},w_{0},\vartheta_{x},\vartheta_{y}\right)}^{T}$ being the vector of displacement components at the mid-surface of the shell panel. Then, the strain–displacement relations based on the coordinate system $\left(x,\theta ,z\right)$ give
(11)$$\left\{\begin{array}{c} \varepsilon_{xx}\\ \varepsilon_{\theta \theta }\\ 2\varepsilon_{x\theta } \end{array}\right\}=\varepsilon =\left\{\begin{array}{c} \frac{\partial u_{0}}{\partial x}\\ \frac{1}{R}\frac{\partial v_{0}}{\partial \theta }+\frac{u_{0}sin\alpha }{R}+\frac{w_{0}cos\alpha }{R}\\ \frac{1}{R}\frac{\partial u_{0}}{\partial \theta }+\frac{\partial v_{0}}{\partial x}-\frac{v_{0}sin\alpha }{R} \end{array}\right\}+z\cdot \left\{\begin{array}{c} \frac{\partial \vartheta_{x}}{\partial x}\\ \frac{1}{R}\frac{\partial \vartheta_{y}}{\partial \theta }+\frac{\vartheta_{x}sin\alpha }{R}\\ \frac{1}{R}\frac{\partial \vartheta_{x}}{\partial \theta }+\frac{\partial \vartheta_{y}}{\partial x}-\frac{\vartheta_{y}sin\alpha }{R} \end{array}\right\}=Hd$$
(12)$$\left\{\begin{array}{c} \gamma_{\theta z}\\ \gamma_{xz} \end{array}\right\}=\gamma =\left\{\begin{array}{c} \vartheta_{y}+\frac{1}{R}\frac{\partial w_{0}}{\partial \theta }-\frac{v_{0}}{R}cos\alpha \\ \vartheta_{x}+\frac{\partial w_{0}}{\partial x} \end{array}\right\}=H_{s}d$$

Here, H and $H_{s}$ are $\left(3\times 5\right)$ and $\left(2\times 5\right)$ gradient-like matrices defined by
(13)$$H=\left[\begin{array}{ccccc} H_{x} & 0 & 0 & z\cdot H_{x} & 0\\ RS & H_{\theta } & RC & z\cdot RS & z\cdot H_{\theta }\\ H_{\theta } & \left(H_{x}-RS\right) & 0 & z\cdot H_{\theta } & z\cdot \left(H_{x}-RS\right) \end{array}\right]$$
(14)$$H_{s}=\left[\begin{array}{ccccc} 0 & -RC & H_{\theta } & 0 & 1\\ 0 & 0 & H_{x} & 1 & 0 \end{array}\right]$$
with $H_{x}=\partial /\partial x,\;H_{\theta }=\partial /R\partial \theta ,\;RS=sin\alpha /R$ and RC=cosα/R. Then, the constitutive equations are expressed as
(15)$$\left\{\begin{array}{c} \sigma_{xx}\\ \sigma_{\theta \theta }\\ \sigma_{x\theta } \end{array}\right\}=\sigma =\left[\begin{array}{ccc} Q_{11} & Q_{12} & 0\\ Q_{12} & Q_{22} & 0\\ 0 & 0 & Q_{66} \end{array}\right]\left\{\begin{array}{c} \varepsilon_{xx}\\ \varepsilon_{\theta \theta }\\ 2\varepsilon_{x\theta } \end{array}\right\}=DHd$$
(16)$$\left\{\begin{array}{c} \tau_{\theta_{z}}\\ \tau_{xz} \end{array}\right\}=\tau =\left[\begin{array}{cc} Q_{44} & 0\\ 0 & Q_{55} \end{array}\right]\left\{\begin{array}{c} \gamma_{\theta z}\\ \gamma_{xz} \end{array}\right\}=D_{s}H_{s}d$$
with $Q_{11}=E_{11}/\Delta ,\;Q_{22}=E_{22}/\Delta ,\;Q_{12}=\nu_{21}E_{11}/\Delta ,\;Q_{66}=G_{12},\;Q_{44}=G_{23},\;Q_{55}=G_{13}$ and $\Delta =1-\nu_{12}\nu_{21}$.

## 3. Analysis of Free-Vibration Using 2-D NEM

For the free vibration approximation of the FG-CNTRC conical shell panel by 2-D NEM, the mid-surface ϖ is discretized into a finite number of nodes and Delaunay triangles, as shown in Figure 2. A coordinate $\left(x,s\right)$ is adopted to identify the position on the mid-surface using the relation of s=Rθ, and Laplace interpolation functions $\psi_{J}\left(x,s\right)$ [25,26] are assigned to each node J. Letting the total number of nodes be N, the NEM approximation $u^{h}\left(x,s,z\right)$ is expressed as
(17)$${\left\{\begin{array}{c} u^{h}\\ v^{h}\\ w^{h} \end{array}\right\}}_{\left(x,s,z\right)}=\sum_{J=1}^{N}{\left\{\begin{array}{c} u_{0}\\ v_{0}\\ w_{0} \end{array}\right\}}_{J}\psi_{J}\left(x,s\right)+\sum_{J=1}^{N}z\cdot {\left\{\begin{array}{c} \vartheta_{x}\\ \vartheta_{y}\\ 0 \end{array}\right\}}_{J}\psi_{J}\left(x,s\right)$$
with $d_{J}={\left(u_{0},v_{0},w_{0},\vartheta_{x},\vartheta_{y}\right)}_{J}^{T}$ being the nodal vector of displacement components at node J.

Referring to the references [25,26], the definition of the Laplace interpolation function and its computation on the conical surface is complex and painstaking. To overcome this difficulty, the physical NEM grid ${ℑ}_{C}=\left[0,\ell \right]\times \left[-s_{L},s_{R}\right]$ with M triangles on the conical surface is mapped to the computational NEM grid ${ℑ}_{R}=\left[0,\ell \right]\times \left[-\theta_{0}/2,\theta_{0}/2\right]$ on the rectangular plane according to the inverse of the geometry transformation $T_{C}^{}$ defined by
(18)$$T_{C}^{}:\;\left(\zeta_{1},\zeta_{2}\right)\in {ℑ}_{R}\;\to \;\left(x,s\right)\in {ℑ}_{C}$$
where
(19)$$x=\zeta_{1},\;s=\left(R_{0}+\zeta_{1}sin\;\alpha \right)\;\zeta_{2}$$
Then, Laplace interpolation functions $\psi_{J}\left(x,s\right)$ are mapped to $\varphi_{J}\left(\zeta_{1},\zeta_{2}\right)$, and one can obtain the inverse Jacobi matrix $J^{-1}$ given by
(20)$$J^{-1}=\left[\begin{array}{cc} \partial \zeta_{1}/\partial x & \partial \zeta_{1}/\partial s\\ \partial \zeta_{2}/\partial x & \partial \zeta_{2}/\partial s \end{array}\right]=\frac{1}{R\left(\zeta_{1}\right)}\left[\begin{array}{cc} R\left(\zeta_{1}\right) & 0\\ -\zeta_{2}\cdot sin\;\alpha & 1 \end{array}\right]$$
for the above geometry transformation. Moreover, the partial derivatives $H_{x}$ and $H_{\theta }$ in Equations (13) and (14) are switched to
(21)$$\frac{\partial }{\partial s}=H_{\theta }=\frac{\partial \zeta_{1}}{\partial s}\frac{\partial }{\partial \zeta_{1}}+\frac{\partial \zeta_{2}}{\partial s}\frac{\partial }{\partial \zeta_{2}}=\frac{1}{R\left(\zeta_{1}\right)}\frac{\partial }{\partial \zeta_{2}}=H_{2}$$
(22)$$\frac{\partial }{\partial x}=H_{x}=\frac{\partial \zeta_{1}}{\partial x}\frac{\partial }{\partial \zeta_{1}}+\frac{\partial \zeta_{2}}{\partial x}\frac{\partial }{\partial \zeta_{2}}=\frac{\partial }{\partial \zeta_{1}}-\frac{\zeta_{2}sin\;\alpha }{R\left(\zeta_{1}\right)}\cdot \frac{\partial }{\partial \zeta_{2}}=H_{1}-\zeta_{2}sin\;\alpha \cdot H_{2}$$
according to the chain rule.

**Figure 2 polymers-15-01987-f002:**
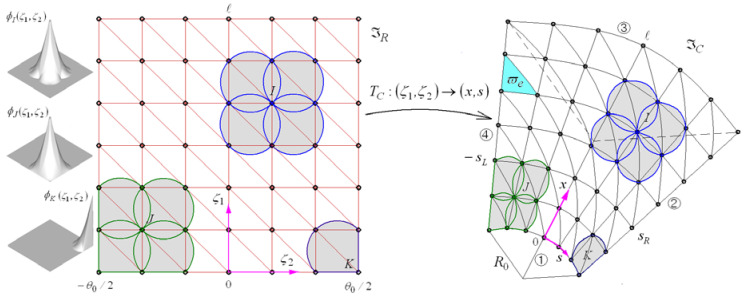
Laplace interpolation functions $\varphi_{J}\left(\zeta_{1},\zeta_{2}\right)$ defined on the rectangular plane and their transformation to $\psi_{J}\left(x,s\right)$ on the conical shell surface.

By substituting Equations (21) and (22) into Equations (13) and (14), one can obtain $\hat{H}$ and ${\hat{H}}_{s}$ in terms of $H_{1}$ and $H_{2}$
(23)$$T_{C}^{-1}:\;H,H_{s}\;\to \;\hat{H},{\hat{H}}_{s}$$
according to the inverse transformation $T_{C}^{-1}$. Then, the NEM approximations of the bending–membrane strain ε in Equation (11) and the transverse shear strain γ in Equation (12) end up with
(24)$$\varepsilon^{h}=\sum_{J=1}^{N}\hat{H}\varphi_{J}d_{J}=\sum_{J=1}^{N}B^{J}d_{J}$$
(25)$$\gamma^{h}=\sum_{J=1}^{N}{\hat{H}}_{s}\varphi_{J}d_{J}=\sum_{J=1}^{N}B_{s}^{J}d_{J}$$

The transverse shear strains in Equation (25), which are directly computed from the standard $C^{0}-$displacement approximation, suffer from shear locking [24,28,29]. To suppress the shear locking, the transverse shear strains are interpolated by employing the concept of the 3-node triangular MITC3+shell finite element depicted in Figure 3. Each element $\varpi_{e}$ in the physical NEM grid ${ℑ}_{C}$ is to be mapped to the master triangular element $\hat{\varpi }$. Moreover, the displacement field is re-interpolated within each element
(26)$${\left\{\begin{array}{c} u^{h}\\ v^{h}\\ w^{h} \end{array}\right\}}_{}^{e}=\sum_{K=1}^{3}{\left\{\begin{array}{c} u_{0}\\ v_{0}\\ w_{0} \end{array}\right\}}_{K}^{e}N_{J}\left(\xi ,\eta \right)+\sum_{K=1}^{3}z\cdot {\left\{\begin{array}{c} \vartheta_{x}\\ \vartheta_{y}\\ 0 \end{array}\right\}}_{K}^{e}N_{J}\left(\xi ,\eta \right)$$
with $d_{K}^{e}={\left(u_{0}^{e},v_{0}^{e},w_{0}^{e},\vartheta_{y}^{e},\vartheta_{y}^{e}\right)}_{K}^{T}$ being the local nodal vector within $\varpi_{e}$ and $N_{K}\left(\xi ,\eta \right)$ being the linear triangular FE shape functions [30]. Then, the element-wise transverse shear strains are interpolated as
(27)$${\hat{\gamma }}_{\theta z}^{e}=\frac{2}{3}\left[\gamma_{\theta z}^{\left(B\right)}-\frac{1}{2}\gamma_{xz}^{\left(B\right)}\right]+\frac{1}{2}\left[\gamma_{\theta z}^{\left(C\right)}+\gamma_{xz}^{\left(C\right)}\right]+\frac{\hat{c}}{3}\left(3\eta -1\right)$$
(28)$${\hat{\gamma }}_{xz}^{e}=\frac{2}{3}\left[\gamma_{xz}^{\left(A\right)}-\frac{1}{2}\gamma_{\theta z}^{\left(A\right)}\right]+\frac{1}{2}\left[\gamma_{\theta z}^{\left(C\right)}+\gamma_{xz}^{\left(C\right)}\right]+\frac{\hat{c}}{3}\left(1-3\xi \right)$$
using the values at the typing points and $\hat{c}=\gamma_{\theta z}^{\left(F\right)}-\gamma_{\theta z}^{\left(D\right)}+\gamma_{xz}^{\left(E\right)}-\gamma_{xz}^{\left(F\right)}$. The analytical manipulation of Equations (27) and (28) using Equations (12) and (26), together with the chain rule between two coordinates $\left(s,x\right)$ and $\left(\xi ,\eta \right)$, leads to
(29)$${\hat{\gamma }}_{}^{e}={\hat{B}}_{e}^{}d_{e}$$
with ${\hat{B}}_{e}^{}$ being the $\left(2\times 15\right)$ matrices in terms of ξ,η,z,α and R and $d_{e}=\left\{d_{1}^{e},d_{2}^{e},d_{3}^{e}\right\}$ being the $\left(15\times 1\right)$ element-wise nodal vector.

Meanwhile, for the free vibration analysis of FG-CNTRC conical shell panels, the standard Galerkin weak form can be derived from the dynamic form of energy principle [31]
(30)$$\int_{-h/2}^{h/2}\int_{\varpi }\left[{\left(\delta \varepsilon \right)}^{T}D\varepsilon +{\left(\delta \gamma \right)}^{T}D_{s}\gamma \right]d\varpi dz+\int_{-h/2}^{h/2}\int_{\varpi }{\left(\delta d\right)}^{T}md\;d\varpi \;dz=0$$
where m is the $\left(5\times 5\right)$ symmetric matrix defined by
(31)$$m=\rho \left[\begin{array}{cc} I & m_{1}^{T}\\ m_{1} & m_{2} \end{array}\right],\;m_{1}=\left[\begin{array}{ccc} z & 0 & 0\\ 0 & z & 0 \end{array}\right]$$
with the $\left(3\times 3\right)$ identity matrix I and $m_{2}=diag\left(z^{2},z^{2}\right)$. Substituting Equations (24) and (28), together with the constitutive relations Equations (15) and (16), into Equation (29) results in the modal equations given by
(32)$$\left[\left(K_{\sigma }+\sum_{e=1}^{M}K_{s}^{e}\right)-\omega^{2}M\right]\;\bar{d}=0$$
to compute the natural frequencies ${\left\{\omega_{I}\right\}}_{I=1}^{N}$ and the natural modes ${\left\{{\bar{d}}_{I}\right\}}_{I=1}^{N}$. Here, the stiffness and mass matrices are defined by
(33)$$K_{\sigma }=\int_{-h/2}^{h/2}\int_{\varpi }B_{}^{T}DB\;d\varpi dz$$
(34)$$K_{s}^{e}=\int_{-h/2}^{h/2}\int_{\varpi_{e}}{\hat{B}}_{}^{T}{\hat{D}}_{s}{\hat{B}}_{e}\;d\varpi dz$$
(35)$$M=\int_{-h/2}^{h/2}\int_{\varpi }\Phi^{T}m\Phi \;d\varpi dz$$
In which $\bar{d}=\left[d_{1},d_{2},\ldots ,d_{N}\right]$, $B=\left[B_{1},B_{2},\ldots ,B_{N}\right]$, $\Phi =\left[\Phi_{1},\Phi_{2},\ldots ,\Phi_{N}\right]$ with $\Phi_{J}=diag\left[\varphi_{J},\varphi_{J},\varphi_{J},\varphi_{J},\varphi_{J}\right]$ and ${\hat{D}}_{s}$ given by
(36)$${\hat{D}}_{s}=\frac{\kappa }{1+\alpha \cdot {\left(L_{e}/h\right)}^{2}}\left[\begin{array}{cc} G_{23} & 0\\ 0 & G_{13} \end{array}\right]$$
with the shear correction factor κ=5/6, the length $L_{e}$ of the longest sides of Delaunay triangles, and a positive constant α called the shear stabilization parameter [21,24]. The value of 5/6 is taken for the shear correction factor because the displacement field is expressed by the FSDT. The value of α is recommended to be 0.1 [21] and taken through the preliminary experiment. This modification of shear modulus was proposed for the sake of further stabilization of the MITC3 element. The numerical integration of three matrices is performed triangle by triangle, and 7 Gauss integration points are used for $K_{\sigma }$ and M while 1 Gauss point is used for $K_{s}^{e}$. A flowchart for the free vibration analysis by the present 2-D NEM-based method is represented in Figure 4.

Meanwhile, three types of boundary conditions, simply-supported (S), clamped (c) and free, are given, where S and C are given as
(37)$$S:v_{0}=w_{0}=\vartheta_{y}=0$$
(38)$$C:u_{0}=v_{0}=w_{0}=\vartheta_{x}=\vartheta_{y}=0$$
at side ① shown in Figure 2, for example. It is noted that the simply-supported condition (37) implies that the shell panel is movable [32].

## 4. Results and Discussion

### 4.1. Verification

The numerical formulae given in Section 3 were coded in Fortran and integrated into the NEM program [2,26], which was previously developed for plate-like structures. The present method was compared with the other reference methods for two completely clamped isotropic and one FG-CNTRC conical shell panels, which are represented in Figure 5. The geometry dimensions of the first example are $R_{0}=1.0\;m,\;\ell \;=3.0\;m,\;h=0.03\;m$, α=π/6 and $\theta_{0}=\pi /3$, and the elastic modulus, Poisson’s ratio, and the density are E=2.1 GPa, ν=0.3 and $\rho =1150\;kg/m^{3}$, respectively. The natural frequencies were normalized as $\hat{\omega }=\omega \ell^{2}\sqrt{\rho h/D}$ with $D=Eh^{3}/12\left(1-\nu^{2}\right)$ being the flexural rigidity.

The lowest four natural frequencies were computed for different NEM grid densities and recorded in Table 1. Referring to Figure 2, the grid density m×n indicates that the total node numbers are m and m along the $\zeta_{2}$- and $\zeta_{2}-$axes. It is observed that the four normalized natural frequencies uniformly decrease and show a stable convergence with a relative difference $\gamma =\left|\Delta \hat{\omega }\right|/\hat{\omega }\times 100\%$ less than 0.3%. According to this convergence experiment, the NEM grids with equal or similar density to 21×21 were used for the remaining numerical experiments.

In Table 2, the lowest four normalized natural frequencies at the grid density 21×21 are compared with the reference solutions obtained by the numerical methods. Where the term without locking suppression indicates the numerical results obtained by the present method in which the transverse shear stiffness matrix $K_{s}^{e}$ in Equation (34) is obtained using the full integration without using MITC3+shell elements. One can see that big errors occur in the free vibration analysis when the locking is not suppressed, such that the four normalized natural frequencies are remarkably larger than the reference solutions. On the other hand, the present method using the MITC approach shows a good agreement with three reference solutions such that the maximum relative difference is 2.61% at mode 4. Here, Bardell et al. [33] used the *h-p* finite element method, Au and Cheung applied the finite strip method, while Xiang et al. [17] used the element-free *kp-Ritz* method, respectively. These three methods used curved meshes, but the present method uses a 2-D planar grid, which is easier to implement.

The second example is the completely isotopic conical shell panel in which the semi-vertex and sub-tended angles $\alpha ,\theta_{0}$ and the thickness h are smaller than those of the first example. In other words, the second example is close to a clamped very thin plate-like structure. The geometry dimensions are $R_{0}=1.0\;m,\;\ell /s=0.2,\;s/h=1000,$$\alpha =7.5^{0}$ and $\theta_{0}=20^{0}$. The material properties are the same as in the first example. The parameter b in the normalization factor is represented in Figure 1. It is clearly seen that the present method is in good agreement with the other three numerical methods, such that the maximum relative difference is 1.53% at mode 3. However, the case without locking suppression shows big errors such that the four normalized natural frequencies are larger than the reference solutions. However, the errors are shown to be smaller than those in Table 2 because the membrane deformation decreases as the sub-tended angle $\theta_{0}$ becomes smaller. In other words, the shell panel becomes a plate-like structure in reverse proportion to $\theta_{0}$ as represented in Figure 5b, so that the membrane-induced locking becomes less severe. Here, Lim and Liew [35] employed the Ritz method using the *pb*-2 shape functions to overcome the mathematical complexity stemming from the shell geometry, similar to the present method. However, differing from the present method, they integrated the whole shell as a single element using the *pb*-2 shape functions.

It has been justified from Table 2 and Table 3 that the present method does not suffer from shear locking. Moreover, it is worth noting that the present results were obtained using 2-D planar, not 2-D curved, and coarse NEM grids. The detailed comparison showing the superiority of NEM against FEM at coarse grids may be referred to in the previous work [36].

The third example is FG-CNTRC conical shell panels manufactured with poly (methyl methacrylate) (PMMA) reinforced with (10,10) SWCNTs. The material properties of the two materials are presented in Table 4, and the efficiency parameters $\eta_{j}\left(j=1,2,3\right)$ introduced in Equations (5)–(7) for PMMA/CNT are given in Table 5, respectively. These material properties and parameters are referred to the rule of mixture results at the temperature of T=300K, which were reported by Han and Elliott [37] and Shen and Xiang [38].

The geometry dimensions are $R_{0}=0.2\;m,\ell =0.8\;m,\alpha =45^{0}$ and $\theta_{0}=120^{0}$, and the volume fraction $V_{cnt}^{*}$ of CNTs is set by 0.12. The natural frequencies were normalized as $\hat{\omega }=\omega \ell^{2}\sqrt{\rho_{m}h/E_{m}}/\left(2\pi h\right)$ with $\rho_{m}$ and $E_{m}$ being the material properties of PMMA. The fundamental frequencies were computed for three different thickness ratios $R_{0}/h$, four different CNT distribution patterns and two different boundary conditions. In Table 6, the results are compared with the reference solutions, which were obtained by Xiang et al. [17] using the element-free kp-Ritz method, where SSSS and CCCC indicate that all sides ①~④ of the shell panel are simply supported and clamped, respectively. It is found that both methods are in good agreement, such that the maximum relative difference is 4.75% at FG-Λ for $R_{0}/h=25$ and SSSS. From the detailed comparison, it is found that the normalized first frequencies obtained by the present method are, as a whole, higher than those of the Ritz method for SSSS and vice versa for CCCC. In addition, it is seen that the relative differences for SSSS are, as a whole larger than those for CCCC.

### 4.2. Parametric Investigation

Next, the free vibration responses of FG-CNTRC conical shell panels are parametrically investigated with respect to the major parameters. The example shown in Figure 5c is taken without changing the material properties, but the volume fraction $V_{cnt}^{*}$ of CNTs and the geometry dimensions of the shell panel are changed depending on the simulation case. The effect of the thickness ratio $R_{0}/h$ on ${\hat{\omega }}_{1}$ is firstly investigated. It is seen from Figure 6a,b that ${\hat{\omega }}_{1}$ uniformly increases in proportion to $R_{0}/h$, regardless of the volume fraction and the distribution pattern of CNTs. It is entirely owing to the calibration with the thickness h, so, actually, $\omega_{1}$ decreases with $R_{0}/h$ because the stiffness of the shell panel decreases as the thickness h becomes smaller. Meanwhile, ${\hat{\omega }}_{1}$ uniformly increases with increasing the value of $V_{cnt}^{*}$ because the stiffness increase due to the increase of $V_{cnt}^{*}$ gives rise to more effect on ${\hat{\omega }}_{1}$ than the mass increase. This explanation is manifest from the fact that ${\hat{\omega }}_{1}$ of the non-CNTRC shell panel is much lower. It is found from Figure 6b that ${\hat{\omega }}_{1}$ is also influenced by the CNT distribution pattern such that the magnitude order of ${\hat{\omega }}_{1}$ is FG-X, FG-U, FG-Λ and FG-O. This is because the stiffness of the shell panel is also affected by the CNT distribution pattern.

Figure 7a,b represent the variations of ${\hat{\omega }}_{1}$ to the semi-vertex angle α for different CNT distribution patterns and different boundary conditions, respectively. Where CSCS indicates that two sides ① and ③ are clamped while the other two sides ② and ④ are simply supported, by referring to Figure 2. It is observed that ${\hat{\omega }}_{1}$ uniformly decreases with increasing the value of the semi-vertex angle, regardless of the CNT distribution pattern and the boundary condition. The radius increase of the shell along the x−axis becomes more apparent as α increases, and accordingly, the stiffness of the shell panel decreases in proportion to the semi-vertex angle. Meanwhile, it is found from Figure 7b that ${\hat{\omega }}_{1}$ is remarkably influenced by the boundary condition such that the magnitude order of ${\hat{\omega }}_{1}$ is CCCC, CSCS, SSSS and CFCF. This order is consistent with the order of boundary constraint.

Figure 8a,b represent the variations of ${\hat{\omega }}_{1}$ to the sub-tended angle $\theta_{0}$ for different thickness ratios and different boundary conditions. All the plots in the two figures show noticeable fluctuations with respect to $\theta_{0}$. It was caused by the sensitivity of numerical natural frequency to the modification of the NEM grid owing to the change in the circular shell side length according to the change of $\theta_{0}$. Moreover, this fluctuation is also reported in the numerical results of Xiang et al. [17]. It is seen from Figure 8a that the variation of ${\hat{\omega }}_{1}$ is not remarkable until $\theta_{0}$ decreases to near $30^{0}$, but thereafter ${\hat{\omega }}_{1}$ abruptly goes up in reverse proportion to $\theta_{0}$, regardless of the thickness ratio $R_{0}/h$. Moreover, the remarkable difference in ${\hat{\omega }}_{1}$ between the thickness ratios until near $\theta_{0}=30^{0}$ becomes negligible as $\theta_{0}$ decreases. It is because conical shell panels approach narrow plate-like structures of zero curvature as $\theta_{0}$ tends to zero, so the fundamental frequency increases owing to the increase of structural stiffness. And the calibrated fundamental frequencies of plate-like structures with different thicknesses approach the same limit as the thickness tends to zero [34,39]. The limit value depends on the boundary condition, as represented in Figure 8b, such that it becomes larger in proportion to the constraint strength of boundary condition. It is worth noting that the variation of ${\hat{\omega }}_{1}$ is not sensitive to $\theta_{0}$ for CFCF, because two sides ② and ④ which are relatively long are not constrained.

The free vibration of the FG-CNTRC conical shell panel was investigated with respect to the shell radius $R_{0}$. It is observed from Figure 9a,b that ${\hat{\omega }}_{1}$ shows a uniform decrease with increasing of the value of $R_{0}$, regardless of the CNT distribution pattern and the panel length. It is because the pane stiffness decreases while its total mass increases as the shell radius increases. However, the decreased slope becomes lower in proportion to the shell radius. Regarding the CNT distribution pattern, the magnitude order of ${\hat{\omega }}_{1}$ is the same as in the previous simulation cases. Meanwhile, it is seen from Figure 9b that ${\hat{\omega }}_{1}$ uniformly increases in proportion to ℓ, but which is entirely owing to the calibration with $\ell^{2}$. The non-calibrated fundamental frequency $\omega_{1}$ decreases with increasing the value of ℓ because the mass increase is superior to the stiffness increase as the shell becomes longer.

Next, two types of three-layered symmetric sandwich FG-CNTRC conical shell panels are considered. One is composed of a PMMA homogeneous core and two FG-CNTRC faces, and the other consists of a FG-CNTRC core and two PMMA homogeneous faces. The free vibration response was investigated with respect to the core thickness ratio $h_{c}/h$. It is seen, from Figure 10a for the PMMA homogeneous core, that ${\hat{\omega }}_{1}$ slightly increases with the increase of $h_{c}/h$, but thereafter, it drops remarkably as $h_{c}/h$ tends to unity. It is because the mass decrease is superior for the core increase up to a certain thickness, but a further increase in core thickness gives rise to a remarkable decrease in panel stiffness. This trend is shown to be severer as the semi-vertex angle α increases. On the other hand, the CNTRC core shows a different variation of ${\hat{\omega }}_{1}$ to the core thickness, as represented in Figure 10b. The normalized first frequency ${\hat{\omega }}_{1}$ uniformly increases in proportion to $h_{c}/h$, and the slope of the increase becomes higher with the increase of $h_{c}/h$ and α. It is because the increase of CNTRC core gives rise to the increase of panel stiffness, and this effect becomes more significant as $h_{c}/h$ and α increase.

## 5. Conclusions

The free vibration of functionally graded CNT-reinforced composite (FG-CNTRC) conical shell panels was analyzed by a 2-D numerical method. The numerical method was developed in the framework of the 2-D planar natural element method (NEM) by introducing a geometry transformation between the shell surface and the planar NEM grid and by adopting the MITC3+shell element. The benchmark experiments were performed to validate the developed 2-D NEM-based numerical method. Moreover, the free vibration responses of FG-CNTRC conical shell panels were profoundly investigated with respect to the associated parameters. The numerical results draw the following major observations:The numerical method reliably and effectively solves the free vibration of FG-CNTRC conical shell panels without causing shear locking with the maximum relative difference of less than 5% to the reference solutions, even for coarse and 2-D planar NEM grids.The normalized first frequency ${\hat{\omega }}_{1}$ increases proportional to the CNT volume fraction $V_{CNT}^{*}$, but it uniformly decreases with increasing the values of semi-vertex angle α and shell radius $R_{0}$.To the sub-tended angle $\theta_{0}$, the ${\hat{\omega }}_{1}$ does not show remarkable change except for slight fluctuation until the decrease of $\theta_{0}$ to a certain value. However, a further decrease of $\theta_{0}$ causes the sudden increase of ${\hat{\omega }}_{1}$, except for CFCF. The increased intensity is dependent on the boundary condition, but it is not sensitive to $R_{0}/h$.The ${\hat{\omega }}_{1}$ uniformly increases with thickness h and length ℓ owing to the calibration, but actually, its non-calibrated value $\omega_{1}$ uniformly decreases proportional to these two parameters.In the symmetric sandwich FG-CNTRC conical shell panels, ${\hat{\omega }}_{1}$ for the PMMA core slightly increases and then drops remarkably proportional to $h_{c}/h$, and vice versa for the CNTRC core.

## Figures and Tables

**Figure 3 polymers-15-01987-f003:**
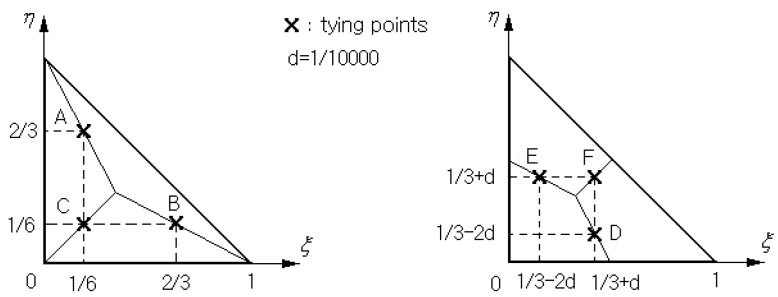
The tying points on the 3-node triangular master element $\hat{\varpi }$ for interpolating the transverse shear strains γ.

**Figure 4 polymers-15-01987-f004:**
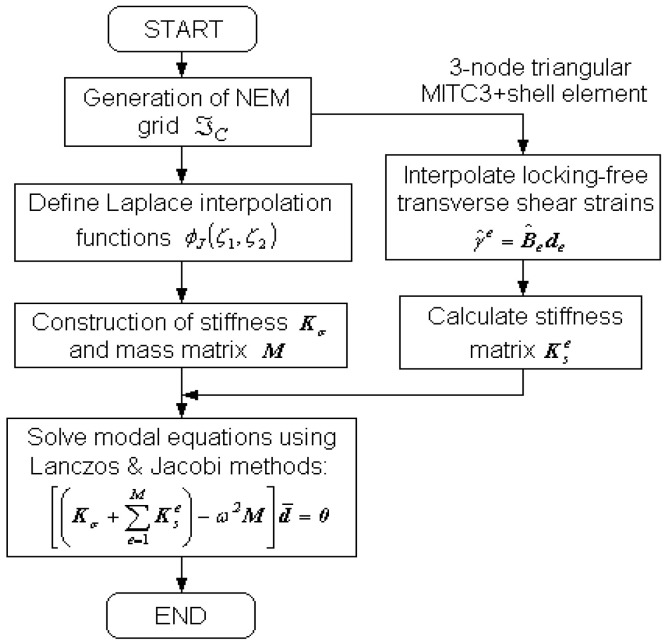
A flow chart for the free vibration analysis by the present 2-D NEM-based locking-free numerical method.

**Figure 5 polymers-15-01987-f005:**
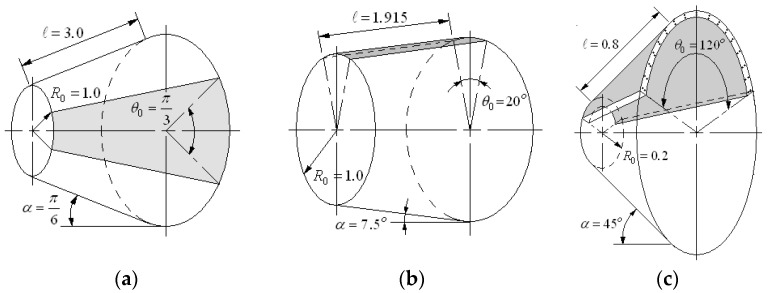
Three conical shell panels taken for numerical experiments: (**a**) isotropic (example 1), (**b**) isotropic (example 2), (**c**) FG-CNTRC (example 3).

**Figure 6 polymers-15-01987-f006:**
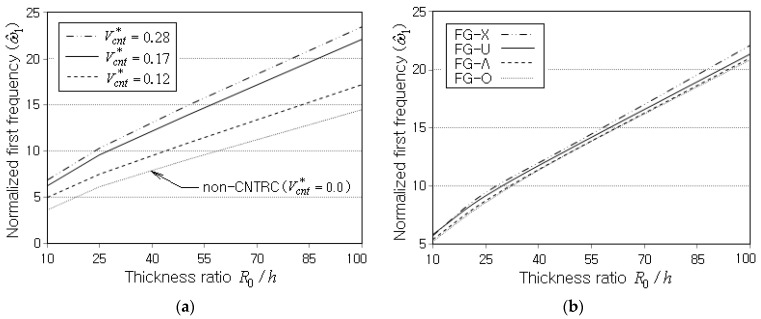
Variation of ${\hat{\omega }}_{1}$ to the thickness ratio $R_{0}/h$ (SSSS, $R_{0}=0.2\;m,\;\alpha =45^{0}$): (**a**) for different CNT volume fractions $V_{cnt}^{*}$ (FG-U), (**b**) for different CNT patterns ($V_{cnt}^{*}=0.17$).

**Figure 7 polymers-15-01987-f007:**
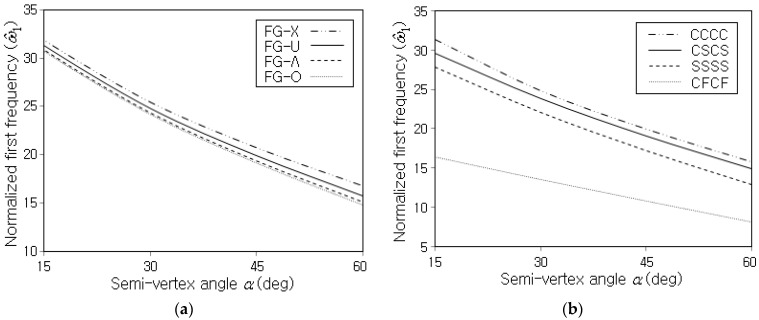
Variation of ${\hat{\omega }}_{1}$ to the semi-vertex angle α ($V_{cnt}^{*}=0.12$, $R_{0}/h=100$): (**a**) for different CNT distribution patterns (CCCC), (**b**) for different boundary conditions (FG-U).

**Figure 8 polymers-15-01987-f008:**
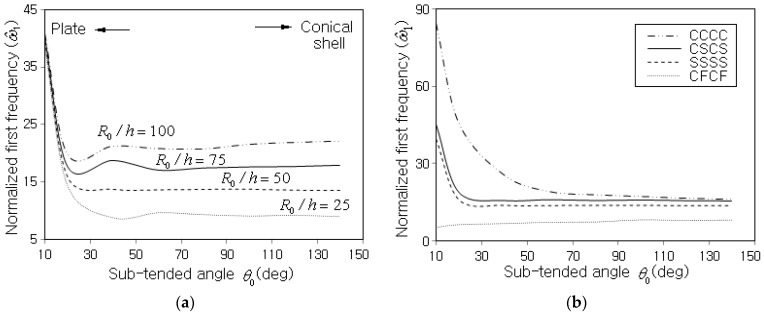
Variation of ${\hat{\omega }}_{1}$ to the sub-tended angle $\theta_{0}$ ($V_{cnt}^{*}=0.17$, $\alpha =45^{o}$, FG-Λ): (**a**) for different thickness ratios (SSSS), (**b**) for different boundary conditions ($R_{0}/h=50$).

**Figure 9 polymers-15-01987-f009:**
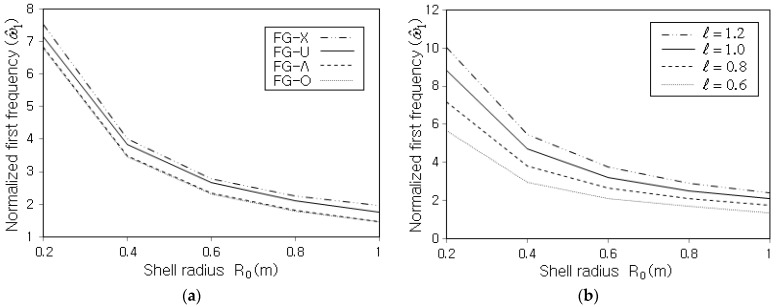
Variation of ${\hat{\omega }}_{1}$ to the shell radius $R_{0}$ (SSSS, $V_{cnt}^{*}=0.12$, $R_{0}/h=25$, $\theta_{0}=120^{o}$, $\alpha =45^{o}$): (**a**) for different CNT distribution patterns (ℓ=0.8 m), (**b**) for different lengths ℓ (FG-U).

**Figure 10 polymers-15-01987-f010:**
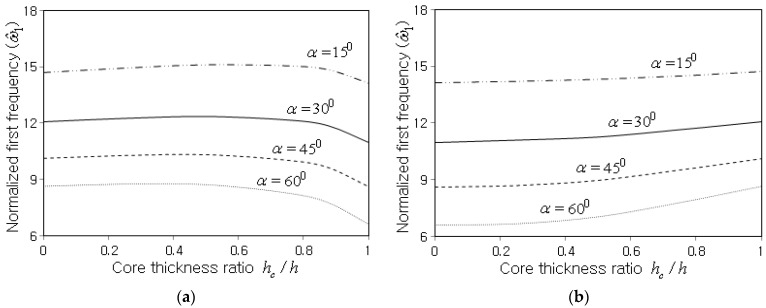
Variation of ${\hat{\omega }}_{1}$ to the core thickness ratio $h_{c}/h$ ($V_{cnt}^{*}=0.12$, FG-U, $R_{0}/h=25$, ℓ=0.8m, $\theta_{0}=120^{o}$, CCCC): (**a**) for the PMMA homogeneous core, (**b**) for the CNTRC core.

**Table 1 polymers-15-01987-t001:** Convergence of normalized natural frequencies $\hat{\omega }=\omega \ell^{2}\sqrt{\rho h/D}$ of the first clamped isotropic conical shell panel to the grid density.

**Mode**	Grid Density					
	13×13	15×15	17×17	19×19	21×21	23×23
1	219.21	215.60	213.55	212.27	211.41	210.80
2	270.25	263.52	260.02	258.14	257.12	256.55
3	321.33	316.65	314.07	312.49	311.44	310.70
4	358.66	353.54	350.94	349.74	349.24	349.06

**Table 2 polymers-15-01987-t002:** Comparison of normalized natural frequencies $\hat{\omega }=\omega \ell^{2}\sqrt{\rho h/D}$ of the first clamped isotropic conical shell panel.

Mode	Bardell et al., 1998 [33]	Au and Cheung, 1996 [34]	Xiang et al., 2021 [17]	Present	Without Locking Suppression
1	209.84	213.4	207.31	211.41	459.36
2	257.11	262.5	255.11	257.12	495.22
3	307.90	314.7	305.62	311.44	555.35
4	351.90	358.6	349.75	349.24	619.71

**Table 3 polymers-15-01987-t003:** Comparison of normalized natural frequencies $\hat{\omega }=\omega b\ell \sqrt{\rho h/D}$ of the second clamped isotropic conical shell panel.

Mode	Bardell et al., 1998 [33]	Xiang et al., 2021 [17]	Lim and Liew, 1995 [35]	Present	Without Locking Suppression
1	235.35	233.39	239.10	236.03	337.67
2	247.45	245.43	251.32	248.84	361.29
3	258.64	256.59	262.61	260.52	397.98
4	269.32	267.23	273.37	269.30	457.46

**Table 4 polymers-15-01987-t004:** Material properties of FG-CNTRC conical shell panels (1,2,3=x,θ,z).

**Materials**	Young’s Moduli (GPa)			Poisson’s Ratios			Shear Moduli (GPa)			Density (kg/m^3^)
	$E_{11}$	$E_{22}$	$E_{33}$	$\nu_{12}$	$\nu_{23}$	$\nu_{31}$	$G_{12}$	$G_{23}$	$G_{31}$	ρ
CNT	5646.6	7080.0	7080.0	0.175	-	-	1944.5	-	-	1400
PMMA	2.5			0.34			0.9328			1150

**Table 5 polymers-15-01987-t005:** The efficiency parameters $\eta_{j}$ for PMMA/CNT for different values of $V_{CNT}^{*}$.

$V_{CNT}^{*}$	$\eta_{1}$	$\eta_{2}$	$\eta_{3}$
0.12	0.137	1.022	0.715
0.17	0.142	1.626	1.138
0.28	0.141	1.585	1.109

**Table 6 polymers-15-01987-t006:** Comparison of normalized first frequencies ${\hat{\omega }}_{1}=\omega \ell^{2}\sqrt{\rho /E}/\left(2\pi h\right)$ for the FG-CNTRC conical shell panels.

Method	B.C.	$R_{0}/h$	CNT Distribution Pattern			
			FG-U	FG-O	FG-X	FG-Λ
Ritz [17]	SSSS	25	7.1582	6.8063	7.5283	6.8247
		50	10.9383	10.6914	11.2019	10.6209
		100	16.6892	16.3410	16.9865	16.3520
	CCCC	25	10.2580	9.3196	11.1090	9.6157
		50	14.0268	13.1407	14.8966	13.3853
		100	19.9495	19.1304	20.8134	19.3277
Present	SSSS	25	7.4946	7.1102	7.4803	7.1495
		50	10.8709	10.4183	11.0825	10.4999
		100	17.2480	16.9044	17.5377	16.9233
	CCCC	25	10.0308	9.1407	10.7619	9.4352
		50	13.6482	12.7137	14.1845	12.9821
		100	19.9310	19.1746	20.7438	19.3556

## Data Availability

Not applicable.

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
