# Peer review of "Free Vibration Responses of Functionally Graded CNT-Reinforced Composite Conical Shell Panels"

_polymers, 2023, doi:10.3390/polym15091987_

Round 1

Reviewer 1 Report

The paper has been devoted to study the vibrational behavior of conical shells reinforced with FG-CNT using 2D numerical method. First-order shear deformation approach has been employed for the displacement field. The general shell theory has been approximated with 2D planar natural element method. Using the MITC3+ approach, the shear locking has been also suppressed. The research method is new and the paper has been provided scientifically. However, some points should be considered by the authors to improve the paper:

1- Some abbreviations used in this article have not been defined for the first time, like: CNT (Carbone Nano-Tube), MITC (Mixed Interpolation of Tensorial Components) and etc. 

2- The main novelty of the paper should be clearly discussed. Using NEM along with MITC scheme is the main novelty?

3- The literature review of the paper is really poor. Many articles have been published on using MITC, vibration of conical shells and using FG-CNT materials. I think the authors should improve this part by dividing it into three sections: Conical shells, numerical method and composite material, especially CNT. The following references can be useful for this literature: https://doi.org/10.12989/scs.2019.30.6.493

https://doi.org/10.12989/scs.2022.43.5.603

4- Is there any relation between the v12 and G12? 

5- Why did the author use the correction shear factor of 5/6?

6- The authors should discuss more about the superiority of the the present solution method compared to the other methods.

7- The reference of value alpha = 0.1 should be presented. 

8- In simply supported shell, please check the freedom of u0. It seems that in this case, the u0 should be also fixed. 

9- Shear locking phenomena has not been controlled in any example. One of the main superiority of the present approach is using the MITC3+ formulation to suppress the shear locking. So this phenomena should be checked.

10- Providing a flowchart to show the detailed steps of the analysis can be useful for other readers. 

  •  

Author Response

Please find attached the response to reviewers' comments (1).

Reviewer 2 Report

A 2D numerical method is developed in the manuscript to simulate the free vibration responses of functionally graded CNT-reinforced composite conical shell panels. The equation derivation and numerical examples are provided for the proposed method. However, the novelty of the research is not clearly presented, and it is suggested to highlight your novelty and give more explanation on your technical improvement and comparison with the published methods. Moreover, the organization of the numerical examples is not clear and there is a lack of pictures for explaining. Kindly please find the following comments.

1.       Kindly please highlight your novelty in the manuscript. What change has already been done? What advantages or improvement by using your method?

2.       Are the FG-CNTRC structures orthotropic or anisotropic?

3.       It is suggested to use subsections in the numerical examples to let the readers to understand the context clearly.

4.       In the numerical examples, different geometry dimensions and different CNT distributions are used. It is suggested to add more figures to illustrate the difference between examples to give the readers a clearer impression of what parameter variations you have investigated.

5.       The simulation results using the proposed method were compared with those from literature. Could you also explain the differences in methodology please? And give the advantages and disadvantage of your methods compared with the published methods.

Author Response

Please find attached the response to reviewers' comments (2).

Round 2

Reviewer 1 Report

Acceptable